# Land-Use, Crop Choice, and Proximity to Ethanol Plants

**Junpyo Park [1], John Anderson [2,\*]**  **and Eric Thompson [2]**

[1]    College of Humanities and Social Sciences, Faculty of Economics, University of Northern Colorado, Greeley, CO 80639, USA

[2]    College of Business, Faculty of Economics, University of Nebraska-Lincoln, Lincoln, NE 68588, USA

\*    Correspondence: janderson4@unl.edu

**Abstract:** This paper examines how proximity to an ethanol plant influences land-use and crop choice among producers. We estimated a Tobit model of crop choice within parcels located in Central Nebraska in a 2014 sample period in order to analyze changes in land-use and crop choice. We employed Geographic Information System (GIS) databases to access relevant data on crop choice and other land uses in the study area parcels, in addition to detailed information on the location and capacity of irrigation wells. We utilized an instrumental variable approach to account for the endogeneity of crop choice with ethanol refinery locations in the study area. Our regional model also took into account specific characteristics of the local processing markets for grains, including animal food manufacturers and livestock as well as ethanol plants. Our estimates revealed that ethanol plants alter land-use in several ways. We found that proximity to an ethanol plant increases the share of land allocated to corn cultivation up to a distance of 30 miles and that the portion of land parcels allocated to corn production falls with distance from an ethanol plant in a non-linear pattern. We also find that land allocation to grassland and pasture rises with distance from ethanol plants.

**Keywords:** land-use; crop choice; ethanol plants; production clusters

---

## 1. Introduction

The renewable fuels sector has been a significant factor in the recent evolution of crop choice in the corn-producing regions of the United States. The U.S. total corn grain production increased to 14,540 million bushels in 2016 from 10,531 million bushels in 2006 [1,2]. Three quarters of that increase was corn production used for ethanol fuel. This resulted from a combination of rising gasoline prices and a suite of state and federal bioenergy policies [3,4]. About five percent of total U.S. transport fuel is comprised of biomass energy, including fuel ethanol and biodiesel in 2016 [5].

The U.S. Department of Agriculture (USDA) reports that the proximity of farmland to ethanol plants has improved and the number of processing facilities has increased [6]. For instance, corn is delivered by trucks for relatively short distances to a nearby ethanol plant and producers consider the transport cost and crop price received. Individual ethanol plants in the midwestern states (IL, IN, IA, KS, KY, MN, MO, NE, OH, SD, and WI) define a market area by the closest plant's input suppliers, farmers, and the density of surrounding corn [7,8].

European research examines how biogas processing impacts agricultural regions, and greenhouse gas (GHG) emissions [9–14]. Biogas production is a significant part of fuel production in European economies [11]. While biogas facilities can operate at the farm level, even farm-level processors draw some feedstock from the surrounding region and larger processing facilities must draw feedstock from throughout the surrounding region [11].

In terms of greenhouse gas (GHG) emissions, one study [15] notes that life-cycle GHG emission reductions are greatest when feedstocks for biofuel plants are grown via perennials on degraded lands not utilized for agriculture, via double cropping, using crop residues and when non-crop feedstocks are utilized such as wood by-products and municipal and industrial waste. While corn-based ethanol plants do not fall into these categories, research on agricultural regions surround these plants can provide insights into the regional concentration of production around processing facilities. It should also be noted that subsidy levels, lags, and uncertainty in refinery development and the treatment of manure inputs and other non-crop feedstocks also influence the potential of biofuel and biogas policy to yield large benefits in reducing greenhouse gas emissions [10,13,16], and influence land prices [12].

The concentration of crop production around biofuel and biogas plants has implications for energy and irrigation policy [17], and also influence water pollution. Clusters of crop and processing activity economize on energy use in hauling bulky crops, by expanding crop production in proximity to the processing plant. One study [11] notes that reducing greenhouse gas emissions from hauling feedstocks is a goal of EU renewable energy policy. Likewise, the concentration of production can lead to higher levels of pesticide and fertilizer application and elevated use of groundwater irrigation. The potential for concentration of feedstock producing crops closest to biofuel or biogas refineries also would have implications for market simulation models that consider feedstock hauling costs [14].

Previous literature on agricultural regions surrounding ethanol plants has utilized USDA's Cropland Data Layer (CDL) to models of crop choice in regions with ethanol plants [18–21]. Some studies use county-level panel data [22] or panel data for other aggregate geographic units [20] to examine how changes in regional ethanol plant capacity influence crop choice. Development of ethanol plant capacity is positively associated with the proportion of acres planted in corn in Iowa counties for 10 years between 1999 and 2009 [22]. Another study [20] utilizes the USDA's CDL data to build identically-sized regions and then examine the effect of a change in the regional production capacity of ethanol plants on corn acres. The study utilizes rail capacity as an instrumental variable for ethanol production capacity within regions and finds evidence that an increase in neighborhood refining capacity is positively associated with corn and total agricultural acres.

Other recent papers examine how proximity to ethanol plants influence crop choice or corn prices [18,23]. One study [18] shows the negative effect of distance to an ethanol plant on corn intensification and extensification. Another study [23] introduces a spatial equilibrium model that estimates the impact on local grain prices from new ethanol plants, mainly in the Midwest regions such as Missouri, Iowa, South Dakota, Illinois, Michigan, and Montana between 2001 and 2002. They find that the advent of twelve new ethanol plants in the regions results in a larger demand for crops, especially corn, and it suggests that the corn price in downstream markets declines with distance from an ethanol plant.

The current study examines the relationship between crop choice and distance to an ethanol plant using parcel-level data for an agricultural region in Nebraska, a corn-producing state located in the Midwest region of the United States. We incorporate alternative sub-county-level data which utilize a resolution of 30 square meters CDL data (compared to 56 square meters [18,21]) to our parcel-level crop choice models. We expand on recent literature by incorporating the instrumental variable techniques developed for regional plant capacity models into the parcel-distance modeling framework [20]. We further examine non-linearity in the relationship between parcel distance and crop choice. Non-linearity could imply that corn cultivation is especially concentrated in the regions closely surrounding ethanol plants. The CDL data also allow for modeling the specific production resources and market characteristics of an individual agricultural region. We control for cropland productivity and groundwater access at the parcel level and for the presence of other local crop processors in the regional market.

## 2. Empirical Model

In our model, we estimated the share of each land parcel which was devoted to the cultivation of corn as a function of distance to the nearest ethanol plant, other local market factors, and land characteristics. Separate equations also were estimated to examine how the share of land parcels devoted to soybean and grassland or pasture was related to these same factors. Specific explanatory variables include distance between each parcel and the nearest ethanol plant, crop productivity [20], and irrigation well capacity [24]. Since transportation costs fall and net prices for corn rise with proximity to an ethanol plant, we anticipated that a profit maximizing producer would increase the share of each parcel devoted to corn production as distance to an ethanol plant or other local processors of corn falls, land would become more productive, and irrigation capacity would rise. These same factors would yield a reduction in the share of each parcel devoted to grassland/pasture. The influence on soybeans would be ambiguous as soybeans are often grown in rotation with corn.

We tested our model in the Central Platte Natural Resources District (CPNRD), a central Nebraska agricultural region with ethanol production facilities and significant corn, soybean, and livestock production. The state of Nebraska is one of the top five agricultural-producing states [25]. Those five states include California, Iowa, Illinois, Minnesota, and Nebraska which represent more than a third of U.S. agricultural outputs and the highest values of crop sales. Corn is among the most important crops produced in the United States. There were approximately 14.4 billion bushels of corn harvested for grain nationwide in 2018 and corn plantings accounted for approximately 28 percent of principal crop acreage planted during that year [26]. Corn accounted for 94 percent of all biofuel production in Nebraska [25]. The increased corn production has required better crop storage facilities which have shifted towards within-farm crop storage and shipping to nearby ethanol plants. Table 1 shows the U.S. ethanol production capacity and production facilities by states in 2016. The state of Iowa has the largest ethanol production capacity and the largest installed ethanol bio-refineries in the United States followed by Nebraska, Illinois, and Minnesota.

**Table 1.** Ethanol production capacity (million gallons/year) and production facilities by state. (Source: 2016 Ethanol Industry Outlook by Renewable Fuels Association) [27].

|  | **Production Capacity** | **Operating Production** | **Installed Ethanol Bio-Refineries** |
|---|---|---|---|
| 1. Iowa | 3947 | 3921 | 44 |
| 2. Nebraska | 2119 | 2066 | 26 |
| 3. Illinois | 1635 | 1597 | 15 |
| 4. Minnesota | 1190 | 1172 | 22 |
| 5. Indiana | 1163 | 1163 | 14 |
| 6. South Dakota | 1032 | 1032 | 15 |
| . | . | . | . |
| . | . | . | . |
| . | . | . | . |
| 15. California | 223 | 218 | 6 |

As indicated in the next section, no corn was grown on 13,120 parcels in our dataset out of a total 31,640 parcels. We therefore utilized a Tobit regression which is an appropriate estimation technique when the value of the dependent variable is censored resulting in many zero values. Using the following equation, we empirically tested the Tobit regression model:

$$S_i = \beta_0 + \beta_1 D_i + \beta_2 D_i^2 + \delta z_i + \gamma X_i + u_i, \tag{1}$$

where the dependent variable $S_i$ is the share of a given land-use in parcel $i$ measured in percent (e.g., corn production) which is bounded between 0 and 1. $D_i$ is a vector of variables that measure the influences of proximity (measured distance in miles) to local markets and $D_i^2$ represents its squared

term in parcel $i$. The term $z_i$ represents a cattle density of the county measured in thousands of cattle on feed per square mile. $X_i$ is a vector of parcel $i$'s land characteristics such as well capacity (measured in the pumping rate of gallons per minute) and crop productivity index (bounded between 0 and 0.99). The term $u_i$ is an unobservable error term.

## 3. Data

We estimated our model using data from the Central Platte Natural Resources District (CPNRD), a central Nebraska agricultural region of 2,136,304 acres. Nebraska is a Midwestern U.S. state with a system of Natural Resource Districts (NRDs) which follow watersheds. CPNRD is a rainwater basin region, one of 23 such NRDs established in Nebraska. Figure 1 displays a satellite imagery gathered by National Agricultural Statistics Service (NASS) on the state of Nebraska. The boundary lines in Figure 1 indicate each Nebraska NRD. Our CPNRD study region is highlighted in cyan in Figure 1.

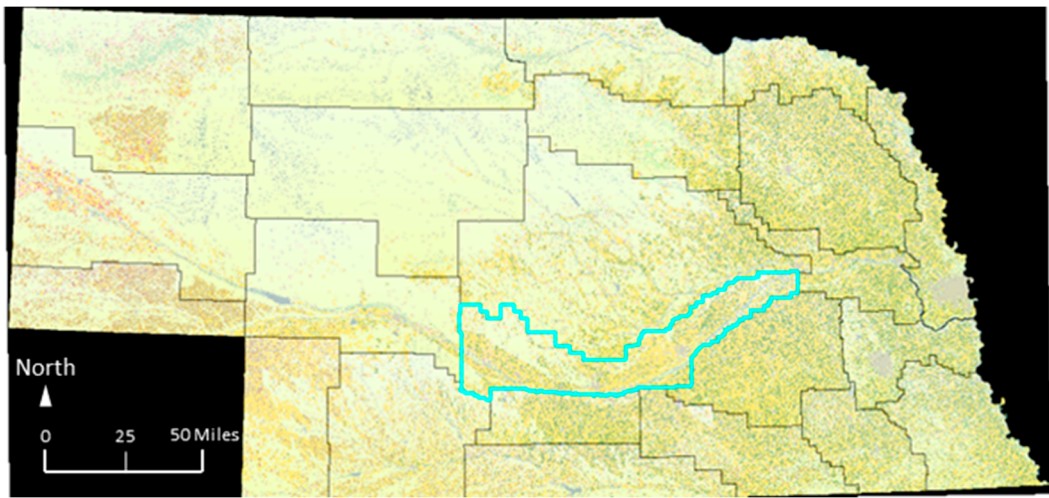

**Figure 1.** Map of study areas (Central Platte Natural Resources District in Nebraska). (Source: USDA Natural Resources Conservation Service: Geospatial Data Gateway, 2014) [28].

Our micro-level dataset included variables for parcel size, ownership, land-use, land characteristics, groundwater access, and proximity to local grain processors in these 11 counties in 2014.[1] We obtained access to the parcel data provided by GIS Workshop, LLC [29]. We restricted our dataset to parcels where less than 99.5 percent of the land area is developed.[2] The number of unique parcels in the study area which meet this criterion is approximately 32,000. The average parcel in our dataset covers 65.57 acres. For our GIS and statistical analyses, we utilized ArcMap version 10.5, Stata version 15.0, and Microsoft Excel. Figure A1 illustrates how we utilized relevant data in our analysis. See Appendix A.

The land-use and land cover data were collected from the Geospatial Data Gateway (GDG) provided by the United States Department of Agriculture (USDA)—Natural Resources Conservation Service (NRCS). Compared to previous studies, our approach using Geographic Information System (GIS) technology provided more detailed information on land-use. Land uses included developed acres, grassland/pasture and acres cultivating corn, soybeans and other crops such as, hay, wheat, sorghum, oats, and potatoes. Therefore, the myriad of land-use alternatives were summarized in five categories: developed areas, uses as grassland or pastureland, corn cultivation, soybean cultivation

---

[1]   There are 11 counties in CPNRD: Buffalo, Custer, Dawson, Frontier, Hall, Hamilton, Howard, Merrick, Nance, Platte, and Polk.

[2]   Otherwise, we would have another 40,000 parcels where developed areas occupy only apartments, mansions, or buildings in the study area.

and cultivation of all other land uses. Land cover data indicated multiple land uses within most of the individual parcels.

Table 2 reports the average share of parcels assigned to each of the five land-use categories in 2014. According to our analysis, the unweighted average for corn cultivation land-use accounted for 24 percent, soybean cultivation was 8.3 percent, grassland/pasture was 33.2 percent, developed area was 17 percent, and other land uses accounted for the rest. Grassland/pasture was the most common landuse followed by corn cultivation. Table 2 also reports the share of individual parcels that include no corn, soybeans, or grassland/pasture. There were a significant number of corn parcels with a zero percent share (13,120 out of 31,640 total), motivating our use of a Tobit model. The bottom half of Table 2 also provides additional detail on the distribution of crop shares in parcels.

**Table 2.** Assigned land-use in parcels.

| Land Use Category | (1) 0% of Acres in Assigned Use | (2) 100% of Acres in Assigned Use | (3) Average (Share of Parcels Assigned to Each Category) | (4) Median (the 50th percentile) |
|---|---|---|---|---|
| Corn | 41.47% | 0.43% | 24% | 1.55% |
| Soybeans | 60.66% | 0.00012% | 8.3% | 0% |
| Grassland/Pasture | 9.37% | 3.26% | 33.2% | 19.12% |
| Developed Area | 18.29% | 0% | 17.1% | 4.50% |
| Others | 26.66% | 1.62% | 17.4% | 3.00% |
| Land Use Category | (5) Between 0.01% and 25% of Acres in Assigned Use | (6) Between 25.01% and 50% of Acres in Assigned Use | (7) Between 50.1% and 75% of Acres in Assigned Use | (8) Between 75.1% and 99.9% of Acres in Assigned Use |
| Corn | 27.04% | 8.12% | 6.41% | 16.54% |
| Soybeans | 27.88% | 4.51% | 2.05% | 4.89% |
| Grassland/Pasture | 44.62% | 14.97% | 13.03% | 14.75% |
| Developed Area | 59.64% | 8.57% | 6.07% | 7.43% |
| Others | 49.78% | 10.21% | 5.81% | 5.93% |

We obtained Gridded Soil Survey Geographic (gSSURGO) information for crop productivity from GDG. The Gridded Soil Survey Geographic data included National Commodity Crop Productivity Index (NCCPI) by USDA-NRCS (Natural Resources Conservation Service). NCCPI uses inherent soil properties, landscape features, and climatic characteristics to assign ratings for dry-land commodity crops such as corn, soybeans, cotton, and small grains. The index ranges from zero (low crop productivity) to 0.99 (high crop productivity). We included the crop productivity index but note that the presence of a nearby ethanol facility could influence both soil quality and landscape features, since producers receiving higher net prices for corn would be expected to invest more in production capacity, including improving soil quality or modifying the farm landscape. As a result, values for the crop productivity index variable could be negatively correlated with distance from an ethanol plant. We note that one other study introduces the time-invariant geographical factor to control for soil quality [20].

Furthermore, we employed water supply capacity [24]. We collected the registered groundwater well data from the Nebraska Department of Natural Resources (DNR). The well data included the number of wells and well capacity, and we are able to project the locations of every single well into ArcGIS for our study. Accordingly, we computed the number of wells and the well capacity in all parcels. We assumed that the number of wells and well capacity were unchanged in 2014.

The locations of all 23 ethanol plants in the state of Nebraska were provided by the Nebraska Ethanol Board. Using the coordinate system, we computed distance from each parcel to the nearest ethanol plant [30]. We also computed distance to the nearest other animal food facility. Data on the location of animal food facilities were gathered [31]. There were too many feedlots to utilize a similar approach for distance to feedlots, and we utilized cattle density, a count of the number (in thousands) of cattle on feed in the parcel's county divided by the land area of the county, as the measure of access. Data on the number of cattle on feed in each county were obtained from USDA-NASS. The cattle

density variable captured supply and demand conditions at the county level; in particular, the supply of land that producers choose to devote to grassland/pasture and the demand for corn production as cattle feed.

Among the 23 ethanol plants in the state of Nebraska, there were three ethanol plants within the CPNRD as well as eight animal food facilities with at least 10 employees. Figure 2 presents the locations of the ethanol plants and animal food facilities as well as the land cover in the study area. In our analysis we included those ethanol plants and animal food facilities with at least 10 employees located in areas adjacent to the CPNRD. Figure 2 provides estimates on acreage for commodities including about 10 to 20 categories out of a possible 85 standardized categories. Lands devoted to corn, soybeans, and grass/pasture are represented in dark yellow, green, and bright ivory, respectively. The figure shows developed lands are scattered throughout the region while corn cultivation is the primary land-use in eastern and central portions of the region and pasture and grassland is the primarily land-use in the west. From Figure 2, it is also evident that many parcels in the region are closer to ethanol plants located outside of the CPNRD, which is why we define our distance variable as the distance to the nearest ethanol plant or other animal food facility, whether or not that plant or facility is located within the CPNRD.

Table 3 shows descriptive statistics for key variables. While the data come from a single agricultural region, there was great variability among individual parcels in terms of land-use, distance to markets, crop productivity and well capacity. Of particular note, the average distance between parcels and ethanol plants is 13.5 miles, with the shortest distance being 0.09 miles and the longest distance 36.15 miles. Also note that since both well capacity and crop productivity are important input factors as farmers consider what crops to plant, we included an interaction term by multiplying the well capacity variable and the crop productivity variable.

**Table 3.** Descriptive statistics.

| Variable | Mean | Std. Dev. | Min | Max |
|---|---|---|---|---|
| Share of corn | 0.240 | 0.345 | 0 | 1.00 |
| Share of soybeans | 0.083 | 0.216 | 0 | 1.00 |
| Share of developed area | 0.170 | 0.263 | 0 | 0.995 |
| Share of grassland/pasture | 0.332 | 0.342 | 0 | 1.00 |
| Distance to nearest ethanol plant (miles) | 13.53 | 6.387 | 0.09 | 36.15 |
| Distance to other animal food facilities (miles) | 16.64 | 12.74 | 0.070 | 44.14 |
| Well capacity (WC) | 34.00 | 800.696 | 0 | 133,088 |
| Crop productivity (CP) | 0.385 | 0.1343 | 0 | 0.8 |
| WC ∗ CP | 13.62 | 289.71 | 0 | 47,468 |
| Number of cattle (thousand) | 122.316 | 79.206 | 27.5 | 285 |
| Size of county (sq. mile) | 815.7 | 410.952 | 439 | 2576 |
| Cattle density (thousands of cattle/sq. mile) | 143.50 | 55.03 | 57.44 | 231.98 |

Note: The data sources collected for our study analysis are as follows. Share of crop choice in parcels—the Geospatial Data Gateway (GDG) and GIS Workshop, LLC [28,29], distance to nearest ethanol plant—location of plants from Nebraska Ethanol Board [32], distance to other animal food facilities—location of facilities from Nebraska Department of Economic Development [31], well capacity—Nebraska Department of Natural Resources [33], crop productivity—USDA Natural Resources Conservation Service [28], number of cattle—USDA National Agricultural Statistics Service [34], size of county—U.S. Census Bureau [35].

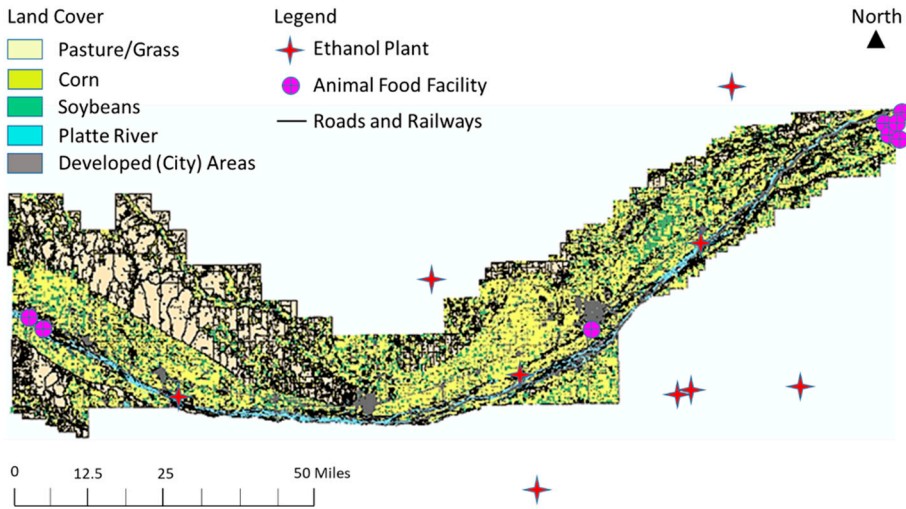

**Figure 2.** Land cover and locations of ethanol plants and animal food facilities. (Source: USDA's Cropland Data Layer—Metadata, 2014) [36].

## 4. Empirical Results

### 4.1. Land Allocated to Corn Cultivation

Table 4 presents results from the Tobit regression model for the share of land parcels allocated to corn cultivation. The first two columns show the results when variables for distance to the nearest ethanol plant are included, but other market characteristics are excluded. Results are presented with and without using an instrumental variable. Of course, the results using instrumental variables are preferred.

**Table 4.** Estimates for effects of local markets on corn planted.

| Variable | Share of Corn Planted in a Parcel | | | |
|---|---|---|---|---|
| | **(1) Tobit** | **(2) Tobit IV** | **(3) Tobit IV** | **(4) Marginal Effects with Tobit IV** |
| Distance to nearest ethanol Plant | −0.0209 *** | −0.0467 ** | −0.0202 *** | −0.0070 *** |
| Distance to nearest ethanol Plant$^2$ | 0.0004 *** | 0.0012 * | 0.0004 ** | 0.0001 * |
| Distance to animal food facility | | | 0.0019 | 0.0008 |
| Distance to animal food facility$^2$ | | | −0.00001 | −0.00002 |
| Well capacity (WC) | −0.0007 *** | −0.0007 *** | −0.0007 *** | −0.0002 *** |
| Crop productivity (CP) | 1.259 *** | 1.2804 *** | 1.2580 *** | 0.5035 *** |
| WC ∗ CP | 0.0021 *** | 0.002 *** | 0.0020 *** | 0.0008 *** |
| Number of cattle/size of county (head/sq. mile) | | | −0.0002 *** | −0.0001 *** |
| Intercept | −0.2210 *** | −0.0633 | −0.2024 *** | |
| Sample size | 31,640 | 31,640 | 31,640 | 31,640 |
| Pseudo $R^2$ (or Log likelihood) | 0.07 | −86,876.4 | −81,542.5 | |
| AR Weak IV Test (Prob > Chi$^2$) | - | 0.02 | 0.003 | |

Notes: The table reports coefficients from Tobit models including IV and marginal effects. *** Significant at the 1 percent level; ** at the 5 percent level; * at the 10 percent level.

Due to high transportation costs, the locations of ethanol refineries are likely to be close to farmlands where corn is grown, which can cause an endogenous correlation between the distance variable and the error term in the model [19,20,37]. To address the endogeneity of locations of ethanol refineries, we calculated the distance between individual parcels and the nearest train station to utilize proximity to the nearest train station as an instrument variable (IV) [38]. Transportation features such as an interstate highway ramp or nearest road intersection are good candidates for valid IVs to address potential endogeneity [39]. Other recent studies [19,20] utilize railroad capacity in a region as an instrument for ethanol plant capacity to address endogeneity and the same approach is applied.

We utilized the IV for distance to an ethanol plant. We conduct Anderson–Rubin (AR) *F*-statistic to examine whether the instrument variable is weak. The test results presented in Columns (2) and (3) of Table 4 indicate that we can reject the null hypothesis that the IV is weak at a 5-percent significance level and conclude that the IV employed is valid to address the endogeneity issue in the study.

Coefficient estimates are similar for the Tobit and Tobit IV models so only results for the Tobit IV model are discussed below. The coefficients of the Tobit model indicate a non-linear relationship between distance to the nearest ethanol plant and the share of a parcel devoted to corn cultivation. The influence of proximity on crop choice declines with increased distance. The coefficient on Distance to Nearest Ethanol Plant is negative and statistically significant but the coefficient on Distance to Nearest Ethanol Plant Squared is positive and significant.

As would be expected, an increase in crop productivity was correlated with the share of a parcel devoted to corn production. While the direct effect of groundwater capacity is negatively associated with the share of a parcel devoted to corn, the interaction of well capacity and crop productivity is positively correlated with the share of parcel acres devoted to cultivating corn.

Column (3) includes variables for the other local market characteristics: the proximity to other animal food facilities and the density of cattle production. Proximity to other animal food facilities had no influence on the share of a parcel devoted to cultivating corn. This may reflect the magnitude of corn use at these facilities. Even though we have focused on animal food facilities with at least 10 employees, corn use at these facilities may be modest relative to ethanol facilities. The density of cattle production in a parcel's county has a negative and statistically significant correlation with the share of corn acreage in parcels. Cattle production represents a local market for corn; however, it also reflects the share of land supplied for use as grassland/pasture. The supply influence is dominant, resulting in the negative correlation. Column (4) shows the marginal effects of each variable.

Figure 3 combines the coefficients for the Distance to Nearest Ethanol Plant and Distance to Nearest Ethanol Plant Squared variables and shows the estimated non-linear relationship between the share of parcel acres allocated to corn production and distance from the nearest ethanol plant. The share allocated to corn production declines with distance up to 30 miles from an ethanol plant. Note that 30 miles is quite close to the maximum distance of 36.15 miles among parcels in our Central Nebraska region.

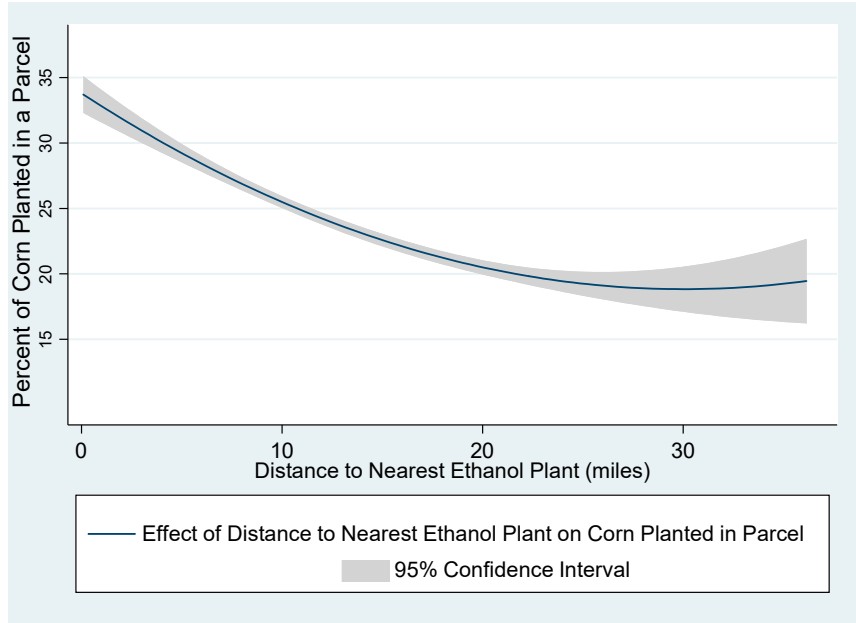

**Figure 3.** Distance to nearest ethanol plant and percent of corn planted in a parcel.

### 4.2. Land Allocated to Soybean and Grassland/Pasture Cultivation

Table 5 presents the results from Tobit IV regression models for the share of land parcels allocated to soybean cultivation or grassland/pasture. Marginal effects also are reported. Critical values for the IV employed in the regression are presented as well. Anderson–Rubin (AR) *F*-statistic in Column (3) for grassland/pasture indicates that nearest distance to a train station is not a weak instrument variable; however, tests indicate it is a weak IV for soybeans.

**Table 5.** Estimates for effects of local markets on soybeans and grassland/pasture.

| Variable | Share of Soybeans Planted in Parcel | | Share of Grassland/Pasture in Parcel | |
|---|---|---|---|---|
| | (1) Tobit IV | (2) Marginal Effects with Tobit IV | (3) Tobit IV | (4) Marginal Effects with Tobit IV |
| Distance to nearest ethanol plant | 0.0037 | −0.0007 | 0.0189 *** | 0.0075 *** |
| Distance to nearest ethanol plant$^2$ | −0.0001 | $8.93 \times 10^{-6}$ | 0.0001 | $-1.72 \times 10^{-6}$ |
| Distance to animal food facility | 0.0045 *** | 0.0007 | 0.005 *** | 0.0018 ** |
| Distance to animal food facility$^2$ | −0.0001 *** | −0.00002 | −0.0001 *** | −0.00004 * |
| Well capacity (WC) | −0.0005 *** | −0.0001 *** | 0.0005 *** | 0.0003 *** |
| Crop productivity (CP) | 0.8757 *** | 0.2524 *** | −0.333 *** | −0.196 *** |
| WC * CP | 0.0014 *** | 0.0004 *** | −0.0015 *** | −0.0009 *** |
| Number of cattle/size of county (head/sq. mile) | 0.00002 | $2.62 \times 10^{-6}$ | 0.0001 *** | 0.0001 *** |
| Intercept | −0.5411 *** | | 0.1718 *** | |
| Sample size | 31,640 | 31,640 | 31,640 | 31,640 |
| Pseudo $R^2$ (or Log likelihood) | −75,347 | | −73,336 | |
| AR Weak IV Test (Prob > Chi$^2$) | 0.53 | | 0.000 | |

Notes: The table reports coefficients from Tobit IV and marginal effects. *** Significant at the 1 percent level; ** at the 5 percent level; * at the 10 percent level.

The share of parcel acreage devoted to soybean production was not associated with distance from nearest ethanol plant. In Columns (1) and (2), the Tobit IV and marginal effect coefficients of distance to animal food facility indicate the share of parcel acreage devoted to soybean production rose with distance from the alternative local market, other animal food facilities. This result may indicate that alternative land uses, such as the cultivation of corn, may be falling with distance from an animal food facility, even though no direct evidence was found for that in Table 4.

As would be expected, an increase in crop productivity was correlated with the share of a parcel devoted to soybeans production. While the effect of groundwater capacity was negatively associated with the share of a parcel devoted to soybeans, the interaction of well capacity and crop productivity had a positive correlation with the share of parcel acres devoted to cultivating soybeans. The density of cattle production had no impact on the share of soybeans acreage in parcels.

In Columns (3) and (4), distance from nearest ethanol plant has a positive and statistically significant relationship with the share of pasture and grassland use in parcels. As distance from parcels to ethanol plants increases, higher transportation costs cause a switch in land-use from planting crops to converting into grassland/pasture. Contrary to the findings in Columns (1) and (2), the growth of well capacity and crop productivity was correlated with a reduction in the percent of a parcel used for grassland/pasture. As expected, the density of cattle production in a parcel's county was positively correlated with the share of grassland/pasture. Counties with a larger share of land devoted to grassland and pasture would have denser cattle production.

### 4.3. Robustness Checks

The results from the previous section show how transport costs can affect land-use decisions by farmers. In this section, we present additional analysis to check whether our main results are robust to alternative specifications and models.

Our baseline analysis included only parcels with less than 99.5 percent of acres in developed areas. In Table 6, we repeat Table 5 analysis using an alternative criterion with only parcels with less than 90

percent of acres in developed acres. This change does not alter the main finding that distance to an ethanol facility plays an important role in influencing land-use.

**Table 6.** Robustness check with Tobit IV for corn, soybeans, and grassland/pasture.

| | 2014 (90%) | | |
| --- | --- | --- | --- |
| | **(1) Tobit IV for Corn** | **(2) Tobit IV for Soybeans** | **(3) Tobit IV for Grassland/Pasture** |
| Distance to nearest ethanol plant | −0.0200 *** | −0.0042 ** | 0.0007 |
| Distance to nearest ethanol plant$^2$ | 0.0003 *** | $6.37 \times 10^{-5}$ | −0.0001 *** |
| Distance to animal food facility | 0.0020 | 0.0055 *** | −0.0051 *** |
| Distance to animal food facility$^2$ | 0.00001 | −0.0002 *** | 0.0001 *** |
| Well capacity (WC) | −0.0006 *** | −0.0005 *** | 0.0002 *** |
| Crop productivity (CP) | 1.277 *** | 0.8890 *** | 0.1699 *** |
| WC ∗ CP | 0.001 *** | 0.0013 *** | −0.0007 *** |
| Number of cattle/size of county (head/sq. mile) | −0.0002 *** | $−5.31 \times 10^{-6}$ *** | 0.00002 |
| Intercept | −0.1940 *** | −0.4712 | 0.0901 *** |
| Sample size | 30,517 | 30,517 | 30,517 |
| Pseudo $R^2$ (or Log likelihood) | −83,002.5 | −77,160.2 | −67,568.5 |
| AR Weak IV Test (Prob > Chi$^2$) | 0.02 | 0.43 | 0.0003 |

Notes: The table reports coefficients from Tobit IV. *** Significant at the 1 percent level; ** at the 5 percent level; * at the 10 percent level.

We also checked the effect of ethanol plant capacity on the change in crop quantity supplied by farmers in the region [20] by adding a variable for the capacity of the nearest ethanol plant. The coefficient on the ethanol plant capacity variable was not significantly different from zero and the included variable did not change other results. Further results are available from the authors, upon request.

### 4.4. Discussion

The expected negative relationship identified in Figure 3 between the share of a parcel devoted to corn cultivation and distance from an ethanol plant has implications for hauling costs for feedstock, and the concentration of irrigation, fertilizer, and pesticide application near ethanol plants. Further, the non-linear relationship identified in Figure 3 and Table 2 indicates even greater concentration in feedstock production in close proximity to ethanol plants. A parcel located at an ethanol plant would be expected to have 33.8% of its territory devoted to corn cultivation versus 29.4% at 5 miles from the plant, 25.5% at 10 miles from the plant and 20.7% at 20 miles from the plant. The cost for hauling all the corn required to fuel an ethanol plant is less in this non-linear case than under a strictly linear curve. The curvature in Figure 3, or analogous curves that may be developed in future research, also can provide a more precise estimate of plant hauling costs in simulation models which include these costs [14].

Results in Figure 3 also have implications for the concentration of fertilizer and pesticide application. Assuming that fertilizer and pesticide applications rises proportionately with land-use, application for corn cultivation would be about 61.2% greater on a parcel located adjacent to an ethanol plant than on a similar size parcel located 20 miles away. Application may even rise by a larger percentage if the growing share of corn cultivation results from a conversion of land from grown corn in rotation with other crops to annual corn cultivation.

A more general implication from our empirical model is to demonstrate that the instrumental variable approach can be utilized in models of the relationship between land-use and distance to ethanol plants, and perhaps also to related questions such as land values and distance. Future research can utilize instrumental variables to test for a negative and non-linear relationship in other regional settings beyond our case study in central Nebraska. Other considerations include whether such non-linear

relationships are present for parcels surrounding other facilities which process food commodities, such as soy-based diesel, or ethanol and biogas plants which utilize crop residue as a feedstock.

## 5. Conclusions and Policy Implications

This paper examines how proximity to ethanol plants and other local crop processors influences land-use and cropping patterns. The results indicate that the location of ethanol plants influences land-use and cropping patterns. We utilized IV techniques to address the potential endogeneity in the location of ethanol plants and we allowed for a non-linear relationship between distance from an ethanol facility and observed land-use. Our analysis relied on measures of land-use based on GIS databases, mitigating the need for producer surveys, and lowering the cost of developing a model which reflected the unique local processing market of a particular region. We employed data from a central Nebraska region with a concentration of grain production as well as ethanol plants, animal food manufacturers, and cattle on feed.

Estimates of the agricultural land-use model show that the broader local processing sector influences land-use and corn and soybean production within the region. In particular, proximity to an ethanol plant increased the share of land allocated to corn cultivation up to a distance of 30 miles. Results further indicate that the marginal impact of distance declines as distance increases, indicating a tendency to concentrate production near the closest ethanol plants. In other words, ethanol plants do not just increase the need for regional corn production, but the plants also concentrate the increased local production in land parcels adjacent to the plant.

The resulting concentration of production in the vicinity of ethanol plants has several implications for policy. There is a potential for a concentration of water pollution in the vicinity of ethanol plants given the elevated fertilizer input needs of the corn cultivation [20]. However, the concentration of crop production around major processing plants also has implications for energy use. Clusters of crop and processing activity economize on energy use and the resulting pollution externalities in hauling bulky crops, by concentrating the required corn cultivation expanding around the processing plant.

**Author Contributions:** Co-author's contributed to individual portions of the research as follows: conceptualization, J.P., J.A., and E.T.; methodology, J.P., J.A., and E.T.; software, J.P.; validation, J.P., J.A., and E.T.; formal analysis, J.P.; investigation, J.P.; resources, J.P., J.A., and E.T.; data curation, J.P.; writing—original draft preparation, J.P, J.A., and E.T.; writing—review and editing, J.P., J.A., and E.T.; visualization, J.P.; supervision, J.A. and E.T.; project administration, E.T.; funding acquisition, E.T.

**Funding:** This research was funded by the Water Sustainability and Climate program (NSF and USDA-NIFA) through a grant made by the USDA (USDA NIFA 2014-67003-22072).

**Acknowledgments:** The authors would like to thank the participants in the Association for University Business and Economic Research 2017 Fall meeting for comments on a draft of this manuscript.

**Conflicts of Interest:** The authors declare no conflict of interest. The funders had no role in the design of the study; in the collection, analyses, or interpretation of data; in the writing of the manuscript, or in the decision to publish the results.

## Appendix A

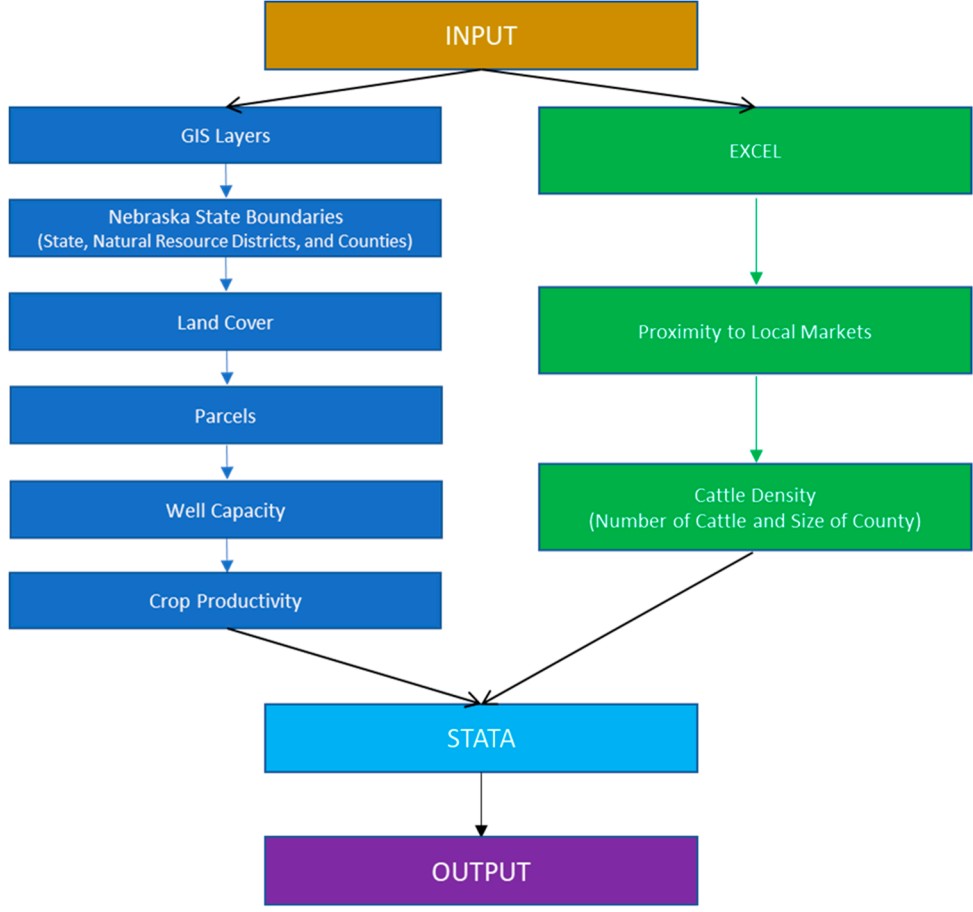

**Figure A1.** Data Flowchart.

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
