# Peer review of "Land-Use, Crop Choice, and Proximity to Ethanol Plants"

_land, doi:10.3390/land8080118_

Round 1

Reviewer 1 Report

The authors used regression models on published datasets to evaluate the effects of distance from ethanol plants in predicting crop choice for farms in Central Platte Natural Resources District of Nebraska.  The methodological approach appears sound, but to a large extent replicates previous analysis by Motamed, McPhail, and Williams (2016), a paper cited extensively in this one.  The novel aspects of the research presented here are that the analysis was conducted at the parcel level, and that non-linearity was considered.  The science is reasonable but in my view, the results are incremental and would be better presented as part of a larger study rather than as a stand-alone paper.

The most important findings from the analysis were that percent parcel cover by corn was higher closer to ethanol plants and that the effects of distance were weaker further from the plants.  This result makes sense and it is useful to have confirmation that the intuitively expected result is borne out.  These results are also consistent with previous findings. 

The authors also note an inverse relationship for percent cover as grass/pasture, which is also related with the density of cattle in the county.  This result also makes intuitive sense, but the interpretation provided is a little backward.  It is suggested that the presence of more cattle result in local demand for pasture and grass.  A more reasonable interpretation of this relationship would be that counties with more farm land area dedicated to pasture are more likely to have more cattle: once a farmer has decided that pasture is the best use of a portion of his or her land area it is natural that livestock would be brought in graze the pasture.

The main body of text is well-written, but the organization of information is unexpected:

* The second half of the Introduction is dedicated to description of the present methodology; I would instead use the Introduction to focus on background with only a short description of approach in the final paragraph, and move further description to the Methods section(s).

* The Literature Review section would be better incorporated into the Introduction.

* The tables and figures are described in the main text; normally only the take-home result of a table would be described in the main text, while the tables themselves would be self-contained; any descriptions can go in figure captions or table footnotes.

Detailed comments follow:

Lines 73-77 of the Inroduction seem to be reporting results; I think this should wait until the Results/Discussion

Lines 78-81 give a roadmap of paper sections; this seems unnecessary.

Figure 1 – Are all mapped areas part of the study region?  Does the cyan highlighting mean anything?  What are the boundary lines drawn (text says there are 11 counties in the region, and there are more than 11 polygons demarkated by these boundaries

Line 156:

Table 2 – main text says it shows parcels assigned to each of five land use categories, but the table only shows three: Corn, Soybeans, Grassland/Pasture

Line 166 – statements like "GIS technology is used to collect information on..." are unnecessary.  On the other hand, it is important to indicate what software and version were used for the analysis, both for the GIS analysis and for the statistical analysis, and I am unable to find that information anywhere in the manuscript.

Figure 2 – what are the black lines and yellow points (or cells?) on the map?  I can't tell what is going on!

Lines 194-208 – is this level of detail really needed for every variable?  the information is summarized in Table 3, and from my perspective it would be better to just provide the most salient information in the text. (Data Section?)

Line 201 – cattle density seems like a more relevant variable than number of cattle in the county.  Good that the average value is included in the text, but can you add this number to the table?

Lines 255-257 – I think there is a "chicken and egg" problem with the idea that cattle create a demand for grassland and pasture. If pasture is deemed by a farmer to be the best use of land, he or she will of course put livestock on it.  It seems more likely to me that the existence of pasture results in a higher presence of cattle than the other way around.  Note, also, that whether cattle create local demand for grassland/pasture will strongly depend on the type of cattle operation and how the animals are managed.  A conventional dairy or CAFO is unlikely to result in increased "demand" for pasture.  Are data available to help draw this distinction?

Lines 258-260 – again, these describe the Figure, rather than reporting the take-home message and letting the figure be self-contained

Line 313 – remove the word "clearly"

p { margin-bottom: 0.1in; line-height: 120%; }

Author Response

Please see attached responses

Reviewer 2 Report

Journal: Land (ISSN 2073-445X)

Manuscript ID: land-543171

Type: Article

Number of Pages: 25

Title: Land-Use, Crop Choice, and Proximity to Ethanol Plants

Overall comments: I had the opportunity to review the manuscript "Land-Use, Crop Choice, and Proximity to Ethanol Plants". This paper is well-written and scientifically very interesting. The results you introduce are well demonstrated by sound data and analysis. Still, few improvements are needed for this paper, especially on the discussion part which is actually missing from the current manuscript. A discussion part would strengthen the applicability and the usability of your models and findings.

The file I had contained 12 pages, not 25.  It is normal?

Line 33: Please, rectify “Department of Agruciulture” with Department of Agriculture

Line 37: It might be useful to give the area estimations (square kilometers or just a distance) from the references of Gallagher and Van Wart, in order to have a first comparison to your outputs.

Line 70: You must add the source and database within the title of Table 1

Line 72-77: I recommend to be even more concise at this stage of the introduction. We already have these information in the abstract and it is not necessary to repeat it here.

Line 85: “changes […] influence”

Line 125-128: I suggest to be more specific with figures. We would appreciate to know how many parcels are concerned by “no corn is grown on many of the parcels” and “all lands”. How many parcels do you have in total in this region? Giving this figure will strengthen your choice of Tobit regression.

Line 129-133: Please, provide the units of your terms

Line 140: Figure 1: This map really needs to be improved. You should add a caption (currently, we do not see or understand the meaning of these colours/labels), scale, north, locations names (of the counties directly on the map) and again, the source of the data layer (USDA…). Also, a general location of you study area within the United States might be useful for non-American readers.

Foot note 2: GIS Workshop, LLC, please provide a reference for this.

Line 148: Please, provide the name of the GIS-software you used, the version and the name of the developer (you mention it later, but it must be specified the first time you discuss GIS technologies)

Line 150: You already said it line 107-108. Remove one of them.

Line 161: again, it would be relevant to give a number or a % of parcels “with a zero percent share”

Line 207-208: Could you be more specific about how you included this interaction term?

Foot notes: I recommend not to use too many foot notes, it interrupts the reading. You can add these notes in the manuscript.

Line 202: Add the data sources you used in the title

Line 210: Figure 2: as recommended for figure 1, a caption should be added to help the reader to understand the meaning of the colours/labels. Add a scale and north as well. In addition, since the data are vector data, there must be a way to improve the quality/resolution of the image during the exportation from ArcGis. I would change the colours in order to increase the visibility of plants and the different landuses.

3. Data: Overall, I suggest to add a general flowchart since you used many relevant data. This flowchart should show the GIS layers, indexes and how you stacked these information.

Line 227: Wouldn’t it be more accurate to calculate the distance to the nearest train station instead of the “nearest railway track”? You might have a close railway track but a further access to the train station where we actually ship the commodities.

Line 265: very interesting figure which sums up the results

Discussion part: I cannot really see a section where you discuss the results of your models. It is a mandatory stage in order to highlights the limits and the core strengths of your methods, how applicable is your model on other region (i.e, wider scale), and beyond that, the future research to improve it. I think it is the missing part of the paper that need to be added.

The conclusion looks fine.

Line 328-334: for instance, the concentration of water pollution (a very interesting point based on the literature you used and on your results) should be discussed earlier, in the discussion part to underline the usability of the results, that the reader may not see in the present form of the paper.

PS: Do not forget to number the references in the manuscript

PS: fill Author Contributions, Fundings, Acknowledgments and Conflicts of Interest and check references (manuscript vs list and vice versa).

Author Response

Please see attached responses

Reviewer 3 Report

The assertion about the distances of the plots to the ethanol plants is correct, as well the transport costs can affect the use the land by the farmers, also the case of the density of livestock production has a greater influence on the demand of pastures vs the demand for corn. All are accounting calculations of farmers and administrators of the various ethanol processors as well as livestock feed. Just one question, are the industries considering offering better conditions for buying products for farmers? In this case the change of the soil can be affected by farmers.

Author Response

Please see attached responses

Reviewer 4 Report

I loved reading the paper. It is well written and presents an interesting case study.

However, I finally decided to suggest to reconsider the paper after a "major revision", because author completely missed to cite and consider a valuable part of the literature on the topic that I think must be inserted in the paper.

I feel that the paper could be improved introducing the topic by a wider point-of-view that take into account the effect of all type of biofuel and biogas production on agricultural productions in developed countries. Specifically, I'd like authors to expand their literature review including the European case which is indeed very interesting and presents many contact points with the research submitted to "Land".

I suggest to look at a papere by Demartini and co-authors published for a good literature review:

Demartini, E., Gaviglio, A., Gelati, M., & Cavicchioli, D. (2016). The effect of biogas production on farmland rental prices: empirical evidences from Northern Italy. Energies, 9(11), 965.

Other relevant and recent pubblications that I feel need to be at least mentioned are:

Tilman, D.; Socolow, R.; Foley, J.A.; Hill, J.; Larson, E.; Lynd, L.; Pacala, S.; Reilly, J.; Searchinger, T.; Somerville, C.; et al. Beneficial biofuels—The food, energy and environment trilemma. Science 2009, 325, 270–271.

Sgroi, F.; Foderà, M.; Di Trapani, A.M.; Tudisca, S.; Testa, R. Economic evaluation of biogas plant size utilizing giant reed. Renew. Sustain. Energy Rev. 2015, 49, 403–409.

Mela, G.; Canali, G. How distorting policies can affect energy efficiency and sustainability: The case of biogas production in the Po Valley (Italy). AgBioForum 2014, 16, 194–206.

Bartoli, A.; Cavicchioli, D.; Kremmydas, D.; Rozakis, S.; Olper, A. The Impact of Different Energy Policy Options on Feedstock Price and Land Demand for Maize Silage: The Case of Biogas in Lombardy. Energy Policy 2016, 96, 351–363

Bartolini, F., Gava, O., & Brunori, G. (2017). Biogas and EU's 2020 targets: Evidence from a regional case study in Italy. Energy Policy, 109, 510-519.

Chatalova, L., & Balmann, A. (2017). The hidden costs of renewables promotion: The case of crop-based biogas. Journal of cleaner production, 168, 893-903. 

Bartoli, A., Hamelin, L., Rozakis, S., Borzęcka, M., & Brandão, M. (2019). Coupling economic and GHG emission accounting models to evaluate the sustainability of biogas policies. Renewable and Sustainable Energy Reviews, 106, 133-148.

Author Response

Please see attached responses

Round 2

Reviewer 1 Report

This is much improved!

Reviewer 2 Report

After the first revisions, it appears that the paper is acceptable in present form. The authors have adressessed well the reviewer's comments and significantly improved figures, methodological points and have incorporated a Discussion part (missing in the first draft).

Therefore, the paper seems to be good for publication.

Reviewer 4 Report

Dear authors,

I am glad that you considered my suggestions in reviewing you paper. At this moment I definitely consider you paper adequate for pubblication in Land.